# Identifying Genetic Mutation Status in Patients with Colorectal Cancer Liver Metastases Using Radiomics-Based Machine-Learning Models

**DOI:** 10.3390/cancers15235648

**Published:** 2023-11-29

**Authors:** Nina Wesdorp, Michiel Zeeuw, Delanie van der Meulen, Iris van ‘t Erve, Zuhir Bodalal, Joran Roor, Jan Hein van Waesberghe, Shira Moos, Janneke van den Bergh, Irene Nota, Susan van Dieren, Jaap Stoker, Gerrit Meijer, Rutger-Jan Swijnenburg, Cornelis Punt, Joost Huiskens, Regina Beets-Tan, Remond Fijneman, Henk Marquering, Geert Kazemier

**Affiliations:** 1Department of Surgery, Amsterdam UMC, Vrije Universiteit Amsterdam, 1081 HV Amsterdam, The Netherlands; n.wesdorp@amsterdamumc.nl (N.W.);; 2Cancer Center Amsterdam, 1081 HV Amsterdam, The Netherlands; 3Department of Pathology, The Netherlands Cancer Institute, 1066 CX Amsterdam, The Netherlands; 4Department of Radiology, The Netherlands Cancer Institute, 1066 CX Amsterdam, The Netherlands; 5Department of Health, SAS Institute B.V., 1272 PC Huizen, The Netherlands; 6Department of Radiology and Nuclear Medicine, Amsterdam UMC, Vrije Universiteit Amsterdam, 1081 HV Amsterdam, The Netherlands; 7Department of Surgery, Amsterdam UMC, University of Amsterdam, 1105 AZ Amsterdam, The Netherlands; 8Department of Radiology and Nuclear Medicine, Amsterdam UMC, University of Amsterdam, 1105 AZ Amsterdam, The Netherlands; 9Department of Medical Oncology, Amsterdam UMC, University of Amsterdam, 1105 AZ Amsterdam, The Netherlands; 10Department of Epidemiology, Julius Center for Health Sciences and Primary Care, University Medical Center Utrecht, 3584 CG Utrecht, The Netherlands; 11Department of Biomedical Engineering and Physics, Amsterdam UMC, University of Amsterdam, 1081 HV Amsterdam, The Netherlands

**Keywords:** colorectal cancer, liver metastases, radiomics, CT scan, genetic mutation, KRAS mutation

## Abstract

**Simple Summary:**

For patients with colorectal cancer with liver metastases, it is important to determine the genetic mutations (e.g., KRAS mutations) of the liver metastases. Around 35–45% of patients with colorectal cancer liver metastases (CRLM) have a KRAS mutation, and genetic mutations are used in treatment planning and prognostication. The aim of this study was to assess if KRAS mutations could be identified on CT scans using radiomics. In the discovery cohort of 255 patients, KRAS mutations could be identified with a good accuracy. In the external validation cohort consisting of 129 patients, the radiomics model performed poorly. These results indicate that radiomics might be used to determine genetic mutations such as KRAS, but foremost emphasize the importance of the external validation of radiomics models. External validation is crucial for the assessment of clinical applicability and should be mandatory in all future studies in the field of radiomics.

**Abstract:**

For patients with colorectal cancer liver metastases (CRLM), the genetic mutation status is important in treatment selection and prognostication for survival outcomes. This study aims to investigate the relationship between radiomics imaging features and the genetic mutation status (KRAS mutation versus no mutation) in a large multicenter dataset of patients with CRLM and validate these findings in an external dataset. Patients with initially unresectable CRLM treated with systemic therapy of the randomized controlled CAIRO5 trial (NCT02162563) were included. All CRLM were semi-automatically segmented in pre-treatment CT scans and radiomics features were calculated from these segmentations. Additionally, data from the Netherlands Cancer Institute (NKI) were used for external validation. A total of 255 patients from the CAIRO5 trial were included. Random Forest, Gradient Boosting, Gradient Boosting + LightGBM, and Ensemble machine-learning classifiers showed AUC scores of 0.77 (95%CI 0.62–0.92), 0.77 (95%CI 0.64–0.90), 0.72 (95%CI 0.57–0.87), and 0.86 (95%CI 0.76–0.95) in the internal test set. Validation of the models on the external dataset with 129 patients resulted in AUC scores of 0.47–0.56. Machine-learning models incorporating CT imaging features could identify the genetic mutation status in patients with CRLM with a good accuracy in the internal test set. However, in the external validation set, the models performed poorly. External validation of machine-learning models is crucial for the assessment of clinical applicability and should be mandatory in all future studies in the field of radiomics.

## 1. Introduction

Colorectal cancer (CRC) is the second-leading cause of cancer-related deaths worldwide [1] and almost half of these patients develop colorectal cancer liver metastases (CRLM) [2,3]. Over the past few decades, substantial advances have occurred in personalized treatment, selecting the optimal treatment based on individual patient and tumor characteristics. The addition of targeted agents to chemotherapy increased response rates to around 60% [4,5,6,7,8]. Targeted agents selectively target the molecular pathways of cancer cells, such as the epidermal growth factor receptor (EGFR) pathway that regulates functions related to cell proliferation, growth, and apoptosis [9]. Overexpression of this pathway leads to a more aggressive disease course and poor prognosis in patients with metastasized CRC; therefore, anti-EGFR agents have emerged as a treatment option [7,8]. The genetic mutation status has been shown to be a promising biomarker for patients with CRLM [10]. Approximately 35–45% of patients with CRLM have a KRAS mutation, which is associated with a poorer prognosis than non-mutated (i.e., wild-type) CRLM [11]. A mutation of KRAS has been shown to lead to a poor response to anti-EGFR-targeted agents [12,13,14]. Pre-treatment determination of the genetic mutations status is therefore an important step in the personalized treatment strategy of patients with CRLM [15]. Nonetheless, the current methods of tumor genomic profiling require invasive methods, such as biopsies. Therefore, it is not feasible to determine the mutation status per individual CRLM. Nowadays, the standard care of patients with CRLM involves non-invasive medical imaging, such as computed tomography (CT), at diagnosis and follow-up. This large number of medical images may potentially be used to find non-invasive biomarkers using advanced imaging analytics, such as radiomics [16]. Radiomics is an advanced method to calculate numerous imaging features from medical images that cannot be quantified by the human eye [16,17]. These imaging features can be used in predictive models using machine-learning classifiers [16]. Research in radiomics has evolved exponentially and is increasingly used to predict clinical outcomes in patients with gastrointestinal cancer, including CRLM [18,19,20]. Radiomics is also applied to identify genetic mutations, which is called radiogenomics [17,21]. More broadly, radiogenomics aims to examine the relationship between imaging features and changes in gene expression. A review by Badic et al. reported that genetic mutations, such as KRAS and microsatellite instability, could be identified using radiomics models in patients with CRC in multiple studies [22]. Likewise, Shi et al. were able to classify the genetic mutation status (KRAS/BRAF/NRAS mutation versus wild-type) of patients with CRLM by making use of a neural network based on CT-based radiomics features [23]. Specifically for patients with mutation status heterogeneity between CRLM [24,25], the mutation status determination via the radiogenomics of individual CRLM could be of additional clinical value. The positive findings of previous studies suggest a relationship between the tumor genotype and its imaging features in metastatic CRC. However, other research also emphasized the methodological shortcomings of studies applying radiogenomics in patients with CRC, using small datasets and lacking external validation [23,26]. This study aims to investigate the relationship between radiomics imaging features and the genetic mutation status (KRAS versus wild-type) in a large dataset of patients with CRLM and validate these findings in an external dataset.

## 2. Materials and Methods

### 2.1. Study Population

The study population consisted of patients registered between 2014 and 2019 in the multicenter randomized clinical trial of the Dutch Colorectal Cancer Group (DCCG), CAIRO5 (NCT02162563) [27]. In this trial, patients with initially unresectable liver-only CRLM were randomized between different systemic therapies based on the primary tumor site and genetic mutation status (KRAS/NRAS/BRAF mutation vs. wild-type) [27]. All patients signed a written consent form, also allowing side-studies. In the current study, only patients with KRAS-mutated or wild-type CRLM were included. Patients from the Netherlands Cancer Institute (NKI) were included as an external validation cohort. This cohort comprised a routine retrospective cohort of patients with CRLM. These patients all underwent biopsy with subsequent next-generation sequencing between 2013 and 2020. Baseline contrast-enhanced CT images were collected with a maximum time difference between the biopsy date and scan date of 90 days.

### 2.2. Data Collection

#### 2.2.1. Genetic Mutation Analysis

Tissue DNA mutation analyses for KRAS (exon 2, 3, and 4) mutations were performed in CCKL-accredited laboratories of participating hospitals following national pathologist guidelines. Tissue mutation analyses were performed mostly on DNA isolated from the primary tumor since tissue from metastases was rarely available (90% versus 10%, respectively).

#### 2.2.2. Imaging Data

All CT scans were performed in one of the 66 centers responsible for trial inclusion using imaging protocols of the local hospitals. The current study only included patients with contrast-enhanced abdominal CT scans in the portal venous phase. Acquisition parameters of CT scans are summarized in Appendix A.

### 2.3. Radiomics Analysis

This study utilized the Radiomics Quality Score (Appendix A) [17].

#### 2.3.1. Tumor Segmentation

All CRLM were semi-automatically segmented in the pre-treatment CT scans to determine the volume of interest (VOI) using IntelliSpace Portal 9.0^®^ (Philips, Best, The Netherlands), by one of three trained members of the research team (NW, MZ, DvdM). All segmentations performed by the research team were verified and if needed, adjusted by an abdominal radiologist (JHvW, IN, SM, JvdB). All segmented CRLM per patient were combined into one segmentation as one VOI.

#### 2.3.2. Feature Extraction and Feature Selection

Radiomics imaging features were calculated from all segmented tumors combined into one VOI per patient using the open-source Pyradiomics package (v3.0.1) [28]. These features were extracted from the original VOIs in the CT scans and as filtered VOIs on which a wavelet filter was applied to suppress possible noise in the image. All feature extraction settings can be found in Appendix A. In order to transform features to a similar scale, all feature values were standardized. Subsequently, the random forest feature selection method was applied in the training set to define the top predictive features based on relative importance. Using validation within the training set, an optimal importance cut-off point of 0.6 was determined. All features with a relative importance above the cut-off were included as input for the classification models.

#### 2.3.3. Data Pre-Processing and Modeling

Model training and testing were performed on the SAS^®^ Viya Analytical platform. The CAIRO5 patients were divided into training and test sets with an 80:20 ratio. The sets were divided using stratification to ensure the similar distribution of the genetic mutation status in the two sets. Hyperparameter tuning was performed on the training set using 5-fold grid-search cross validation (Appendix A). Three machine-learning classification algorithms were used: Random Forest (RF), Gradient Boosting (GB), and Gradient Boosting + LightGBM (GBM). In addition, the three models were joined into an Ensemble classifier, which was created using a voting averaging method. Figure 1 summarizes all steps of the radiomics workflow.

(1) Patient diagnosed with CRLM. (2) Segmentation of the CRLM to determine the VOI on a CT scan. (3) Radiomics feature extraction from the VOI: shape features, first-order features, and texture features (1 texture before systemic therapy; 2 texture after systemic therapy). (4) Random forest feature selection to determine top predictive features. (5) Development of machine-learning models to determine KRAS status, compared with AUC-ROC.

### 2.4. External Validation

Baseline CT scans in the portal venous phase of patients with CRLM from the NKI were collected. All CRLM were segmented by one of the members of our research team, verified, and if needed, adjusted by one abdominal radiologist (JHvW, IN, SM, JvdB) using using IntelliSpace Portal 9.0^®^ (Philips, Best, The Netherlands)in the same manner as the CAIRO5 cohort. Similar data pre-processing and modeling were applied. In addition, ComBat harmonization was applied to the extracted features of the external validation cohort to reduce the center effect.

### 2.5. Statistics

The discriminative performance of the machine-learning models was assessed using the receiver operating curve (ROC), supported by the area under the receiver operating curve (AUC), accuracy, sensitivity, and specificity measures. The optimal cut-off value was determined by taking the point with the greatest difference between the sensitivity and 1—specificity values in the ROC curve (the Youden index). The 95% confidence intervals (CI) of the AUC scores were determined using the Delong test to give an indication of their robustness. The calibration was evaluated using the locally estimated scatterplot smoothing (LOESS) calibration curve and the Brier score, demonstrating the agreement between the observed and predicted probabilities. Baseline characteristics of the discovery cohort and external validation cohort were compared. The relation between the baseline variables and genetic mutation status was assessed using univariable linear regression models. Categorical variables were reported as frequencies and percentages and compared with Chi-square test or Fisher exact test, as appropriate. Continuous variables were displayed as median with interquartile range (IQR) and compared with Mann–Whitney U-test or *t*-test based on the distribution of data. Tests were considered statistically significant with a two-sided *p*-value < 0.05. Statistical analyses were performed using SAS^®^ Studio (version 5.2, SAS^®^ Viya^®^.03.05, SAS Institute Inc., Cary, NC, USA).

## 3. Results

### 3.1. Study Population

In total, 255 of 368 eligible patients from the CAIRO5 trial were included in the discovery cohort of this study (Figure 2). The 80:20 train/test ratio resulted in a training cohort of 204 patients and a test cohort of 51 patients. Of the 167 eligible patients for external validation, 129 were included (Figure 3).

The median number of CRLM per patient within the discovery cohort was 12 versus 5 in the external validation cohort (*p* < 0.0001) and the median slice thickness within the discovery cohort was 5.0 mm versus 1.0 mm in the external validation cohort (Appendix A). All other baseline characteristics of the discovery and external validation cohort were similar (Table 1). Age (*p* = 0.21), sex (*p* = 0.93), and number of liver metastases (*p* = 0.61) were not significantly associated with the KRAS mutational status in the discovery cohort.

### 3.2. Feature Selection and Modeling

In total, 851 radiomics features were extracted. Using random forest feature selection, 10 features were selected as the input variables for the classification models (Appendix A). The optimal model parameters were found using cross-validated hyperparameter tuning (Appendix A).

### 3.3. Performance of the Models

#### 3.3.1. Discrimination Accuracy

In the test set, the RF, GB, GBM, and Ensemble models reached AUCs of 0.77 (95%CI 0.62–0.93), 0.77 (95%CI 0.64–0.90), 0.72 (95%CI 0.57–0.87), and 0.86 (95%CI 0.76–0.95) with an accuracy of 0.80, 0.73, 0.71, and 0.77, respectively (Figure 4). In the external validation cohort, the RF, GB, GBM, and Ensemble models reached AUCs of 0.54 (95%CI 0.44–0.64), 0.52 (95%CI 0.42–0.62), 0.56 (95%CI 0.46–0.67), and 0.47 (95%CI 0.37–0.56), respectively (Figure 5). An overview of all the evaluation metrics is shown in Table 2.

#### 3.3.2. Calibration

In the test set, the RF, GB, GBM, and Ensemble models reached Brier scores of 0.22, 0.21, 0.24, and 0.15, respectively (Figure 6). In the external validation cohort, the RF, GB, GBM, and Ensemble models reached Brier scores of 0.28, 0.41, 0.41, and 0.44, respectively (Figure 7).

## 4. Discussion

This study demonstrates that the genetic mutation status (KRAS mutation versus no mutation) of patients with CRLM could be identified with a good accuracy in the discovery cohort, using machine-learning models incorporating pre-treatment CT-based imaging features. However, the machine-learning models showed poor AUC scores in the external validation cohort, indicating no relationship between the genetic mutation status and radiomics imaging features.

If radiomics could prove to be predictive of the mutation status on a lesion level, it could facilitate personalized cancer treatment on a lesion level in the future. Brudvik et al. have reported that RAS mutations are associated with a higher positive margin resection rate [29]. Thus, for RAS-mutated CRLM, a more careful intraoperative assessment of the resection margin is recommended [29].

In previous studies, genetically driven physiological functions related to venous invasion in liver cancer were correlated with imaging features, suggesting that biological or morphological changes following genetic mutations are detected on medical imaging with radiomics [30]. But following the results in the external validation cohort of this study, the question remains if there is any relationship between radiomics and genetic mutations in patients with CRLM.

The models’ accuracy in this study’s test set was higher than a similar study, in which an AUC of 0.74 was reached in predicting the RAS/BRAF versus wild-type mutation status in patients with CRLM based on radiomics features [23]. Two other studies predicting the genetic mutation status in patients with colorectal cancer achieved comparable AUC scores of 0.82 and 0.83 [31,32]. However, all of these studies were conducted using smaller datasets and more importantly, the models were not externally validated [23,31] and therefore at risk of overfitting.

External validation of machine-learning models is crucial for the assessment of clinical applicability in daily clinical practice. However, it is a major challenge to develop models that also perform successfully in unknown external datasets [33,34]. Hence, machine-learning models should be trained on data that are as realistic and robust as possible. The variety in CT acquisition parameters could be considered a strength concerning external validity, as it is a good representation of different CT scans in daily practice since the CT scans were acquired from 66 hospitals. However, the developed models still performed poorly in the external validation set. This could imply that the selection of patients used for model development had a large influence on the generalizability of the models, as the population consisted of selected patients with initially unresectable CRLM and a median number of 12 metastases. Additionally, a difference in slice thickness was observed between the discovery cohort and external validation cohort.

In addition to patient selection, tumor segmentation also has a considerable influence on model performance, as this segmentation determines which voxels are analyzed and thus can influence the extracted feature values [17]. In this study, the CRLM were validated by one expert radiologist, which could have introduced interobserver bias [35], and therefore the stability of the radiomics features between different segmentations was not investigated. The inability to reject instable features may have affected the generalizability of the models [17]. The use of fully automatic segmentation models in the future would be valuable in increasing reproducibility and constructing robust models. 

Another limitation is that the models were constructed based on imaging features calculated from the segmentation of all the CRLM combined. As a result, it was assumed that the mutation status was uniform amongst all the CRLM in one patient and intertumor heterogeneity was not taken into account. Two studies have shown intertumoral heterogeneity in 2/97 (2.1%) and 11/94 (11.7%) of patients with multiple CRLM, respectively [24,25]. Only the first study, however, used highly sensitive mutation analyses. Finally, it would be interesting to investigate if radiomics models can distinguish between RAS and BRAF mutations or even between specific KRAS mutation variants. Research has shown that patients with CRLM with a KRAS A146-mutated tumor have a higher tumor burden, associated with worse clinical outcomes, compared with patients with other KRAS mutation variants [36]. Due to high imbalance ratios in terms of both RAS/BRAF mutations and specific KRAS mutation variants in the current study, these analyses could not be performed without a high risk of bias.

In an in-depth analysis to explain the poor performance in the external validation cohort, it was firstly observed that the correlation between the individual features and the outcome was weak. Moreover, the correlation coefficient between the internal train, internal test, and external validation set varied significantly. The most predictive features in the internal test were not predictive in the external validation set. Secondly, the distribution of the KRAS mutation and wild-type was distributed similarly between the internal and external dataset. Both were around half of the patient population, which is an adequate representation of the distribution in the daily patient population. Hence, the misclassified samples were not under-represented in the training data.

We advocate for more standardized and robust radiomics research, as the lack of generalizability has to be addressed before a conclusion can be drawn about the potential clinical value of radiomics. External validation is crucial in testing the generalizability of radiomics models, which many previous studies have lacked. Our study has shown that despite promising internal results, the radiomics models showed an inadequate performance in the external validation cohort. In an attempt to build radiomics models that perform well across different, unseen patient cohorts, it is important to have a standardized approach in their development. This could include multiple patient cohorts in the development phase, and also external validation across multiple patient cohorts.

## 5. Conclusions

In conclusion, machine-learning models incorporating pre-treatment CT imaging features could identify the genetic mutation status (KRAS mutation and no mutation) in patients with CRLM with a good accuracy in the test set. However, in the external validation set, the models performed poorly. This study demonstrates the importance of external validation, which should be mandatory in all future studies in this field. Automation and standardization are equally necessary to further explore the potential of radiomics.

## Figures and Tables

**Figure 1 cancers-15-05648-f001:**
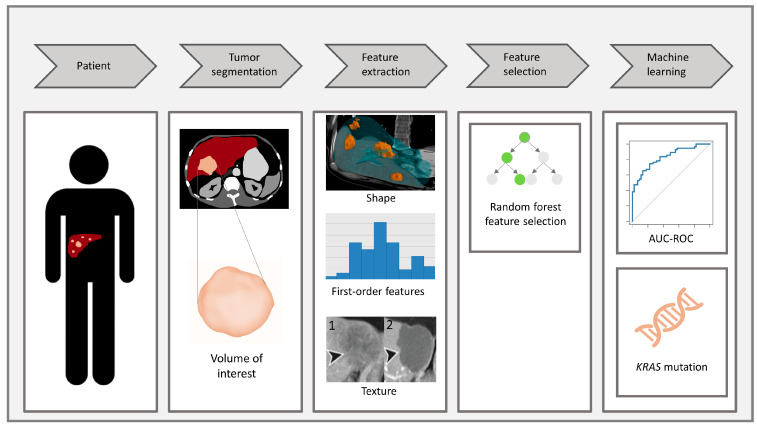
Illustration of the radiomics workflow. Abbreviations: AUC-ROC, area under the receiver operating characteristic curve; CRLM, colorectal liver metastases; CT, computed tomography; VOI, volume of interest.

**Figure 2 cancers-15-05648-f002:**
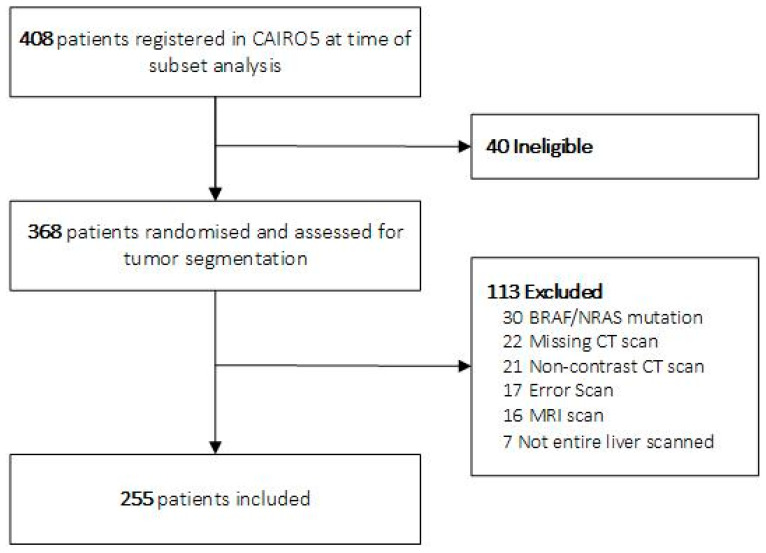
Flowchart of CAIRO5 patient inclusion. Abbreviations: CT, computed tomography; MRI, magnetic resonance imaging.

**Figure 3 cancers-15-05648-f003:**
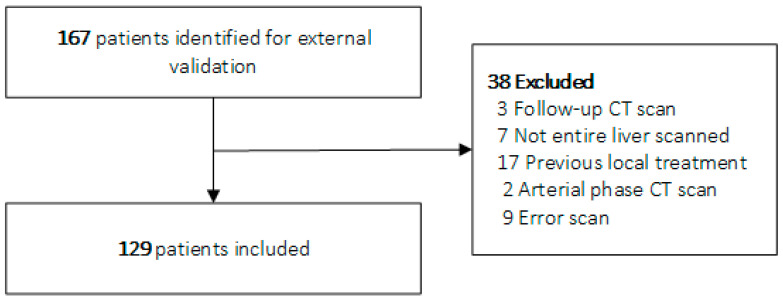
Flowchart of patient inclusion external validation cohort. Abbreviations: CT, computed tomography.

**Figure 4 cancers-15-05648-f004:**
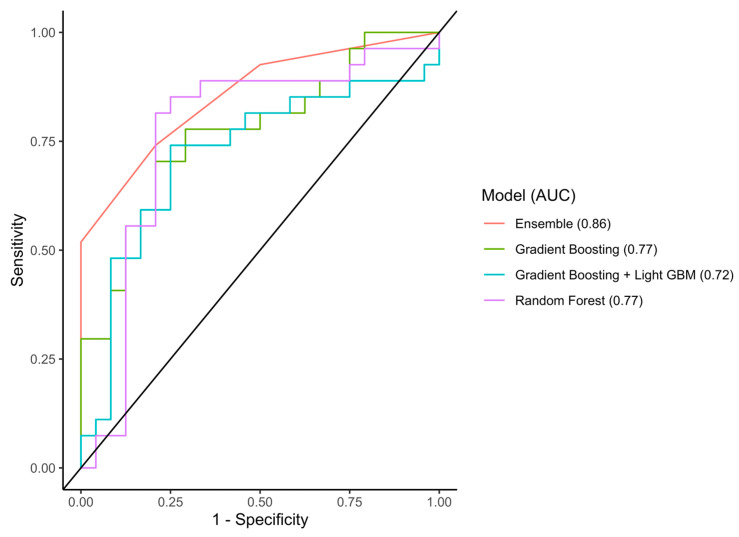
AUC-ROC of the three machine-learning models in the internal test set. Abbreviations: AUC-ROC, area under the receiver operating characteristic curve.

**Figure 5 cancers-15-05648-f005:**
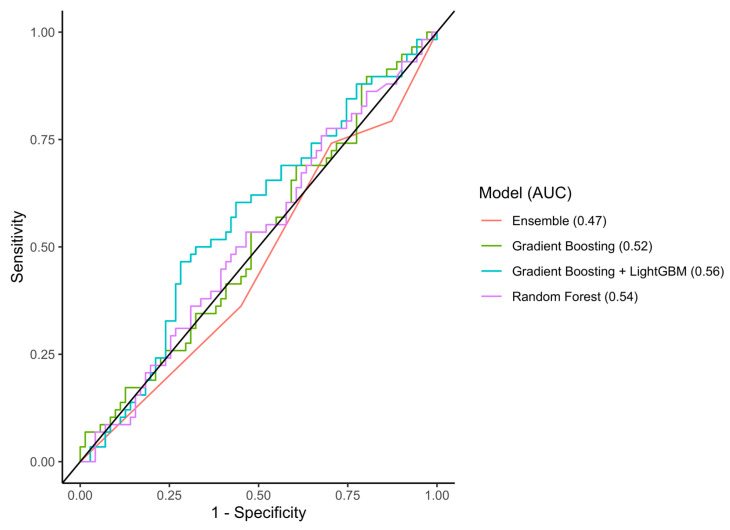
AUC-ROC of the three machine-learning models in the external validation set. Abbreviations: AUC-ROC, area under the receiver operating characteristic curve.

**Figure 6 cancers-15-05648-f006:**
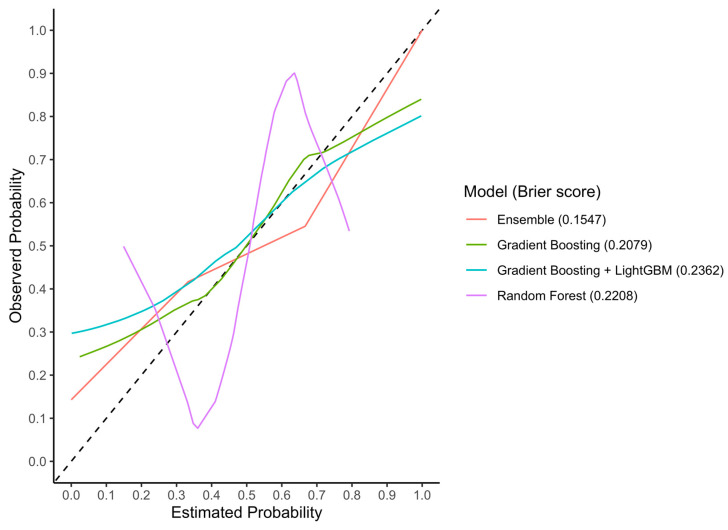
LOESS calibration curves and Brier scores of the three machine-learning models in the internal test set. Abbreviations: LOESS, locally estimated scatterplot smoothing.

**Figure 7 cancers-15-05648-f007:**
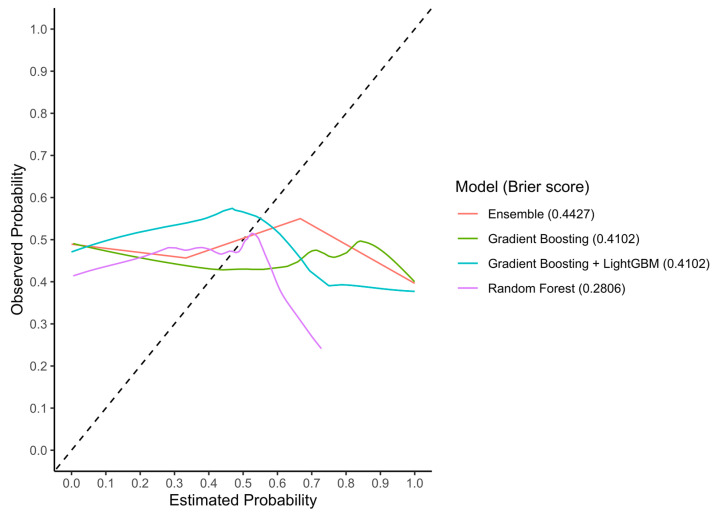
LOESS calibration curves and Brier scores of the three machine-learning models in the external validation set. Abbreviations: LOESS, locally estimated scatterplot smoothing.

**Table 1 cancers-15-05648-t001:** Baseline patient characteristics.

Baseline Characteristics	Discovery Cohort (*n* = 255)	External Validation Set (*n* = 129)	*p*-Value
Age (years)	62 (55–70)	62 (53–69)	0.607
Sex			0.068
Male	170 (66.7)	73 (56.6)	
Female	85 (33.3)	56 (43.4)	
KRAS mutational status			0.149
Mutation	136 (53.3)	58 (45.0)	
Wild-type	119 (46.7)	71 (55.0)	
Number of liver metastases	12 (7–23)	5 (2–11)	<0.0001

**Note**—values are shown as median (interquartile range) or number of participants (percentage).

**Table 2 cancers-15-05648-t002:** Model performance in discriminating KRAS mutant versus wild-type mutation status.

Machine-Learning Classifier	Cohort	AUC (95% CI)	Sensitivity	Specificity	Accuracy
Random Forest	Train	0.97 (0.95–0.99)	0.82	0.95	0.89
	Test	0.77 (0.62–0.93)	0.85	0.75	0.80
	External validation	0.54 (0.44–0.64)	0.44	0.54	0.50
Gradient Boosting	Train	0.96 (0.94–1.00)	0.95	0.98	0.98
	Test	0.77 (0.64–0.90)	0.67	0.79	0.73
	External validation	0.52 (0.42–0.62)	0.46	0.57	0.52
Gradient Boosting (LightGBM)	Train	0.98 (0.97–1.00)	0.98	0.99	0.98
	Test	0.72 (0.57–0.87)	0.75	0.67	0.71
	External validation	0.56 (0.46–0.67)	0.28	0.59	0.42
Ensemble	Train	0.97 (0.94–1.00)	0.97	0.93	0.95
	Test	0.86 (0.76–0.95)	0.93	0.58	0.77
	External validation	0.47 (0.37–0.56)	0.30	0.74	0.50

## Data Availability

We closely collaborate with international partners in the Pancreatobiliary and Hepatic Artificial Intelligence Research (PHAIR) consortium. A PHAIR github will be created to share all results and data.

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
