# Peer review of "Identifying Genetic Mutation Status in Patients with Colorectal Cancer Liver Metastases Using Radiomics-Based Machine-Learning Models"

_cancers, 2023, doi:10.3390/cancers15235648_

Round 1

Reviewer 1 Report

Comments and Suggestions for Authors

In this paper, the authors fitted machine learning models for predicting KRAS mutation status in patients with colorectal cancer liver metastases (CRLM) using radiomics features. They demonstrated that their fitted model could predict KRAS mutation with 0.86 AUC on the internal test set but failed to predict KRAS mutation (0.52 AUC) on an external validation set. The addressed question is interesting and relevant to the bioinformatics community. However, the reviewer has the following concerns:

1. The authors should perform an in-depth error analysis to identify possible reasons for the model’s poor performance on the external validation set and propose potential, more detailed, approaches for facilitating “more robust radiomics research” accordingly. For example, the authors may investigate the importance of different features in the fitted model and whether the most important features are still associated with mutation status in the external data. They may also check if the misclassified samples in the external data are not well-represented in the internal data.

2. Covariates that are potentially predictive of KRAS mutation should be included in the models, especially those significantly different in the internal and external cohorts (e.g., gender, number of liver metastases, median slice thickness).

3. XGBoost and LightGBM are known to perform better than a normal gradient boosting algorithm and outperform random forests in some cases. The authors should include these methods in their analysis in order to provide a more thorough evaluation.

4. The paper only mentioned performing ComBat harmonization to the features of the external validation cohort, whereas it should be applied to both the internal and external datasets for batch effect correction.

5. In Table 2, the authors should (1) provide confidence intervals for the model performance metrics and (2) list training set performances in addition to the testing set performances.

6. Figure 1 (especially the last three columns) should be improved so that the cliparts are less generic and more informative. For instance, the authors may provide concrete examples of different shapes and textures in the extracted features.

Comments on the Quality of English Language

There are minor typos in the manuscript. For example, “radiomcs” should be “radiomics” in line 39.

Reviewer 2 Report

Comments and Suggestions for Authors

In the study, the authors investigated the relationship between radiomics imaging features and genetic mutation status in a large multi-center dataset of patients with CRLM and validate these findings in an external dataset. And it showed the machine learning models incorporating CT imaging features could identify the genetic mutation status in patients with CRLM with good accuracy in the internal test set. But in the external validation cohort performed poorly. So the following questions need to be answered.

(1) In the study population part, Baseline contrast enhanced CT images were collected with a maximum time difference between the biopsy date and scan date of 90 days. Is it possible that the tumor genes may undergo new mutations in 90 days, which will affect the accuracy of the result?

(2) The CT scans parameters were different between external and internal cohort, which can affect the results which showed the external validation cohort performed poorly, because 5mm slice can lost many tumor characters.

(3) What kind of CT scanner machine was used for the study?

(4) The size of a tumor may potentially impact its homogeneity. If the tumor is too large, there might be necrosis in the center of tumor, which can affect the features in the CT images. It is advisable to include tumor size and categorize them based on the tumor size for statistical analysis. 

Reviewer 3 Report

Comments and Suggestions for Authors

The manuscript by Wesdorp et al., entitled “Identifying genetic mutation status in patients with colorectal cancer liver metastases using radiomics based machine learning models” the manuscript seems to be interesting and the whole analysis is well done, and the results are very existing. However, I am very curious about this study as a guideline for the diagnostic and the treatment of this type of tumor. My mean concern about this study is the limitation of the analysis of big dataset that can only offer a general concept rather than specific concept. Thus based the tumor heterogeneity, the phenomena that are common different tumor types. The authors have already analyzed and evaluated the data set without considering the role tumor heterogeneity in the whole analysis.  However, the manuscript is well written, the study is well designed, and the data are logical and clear, the validity of the data as guideline for the treatment and diagnostic and even the assessment of treatment prognosis is very poor

Many thanks

Round 2

Reviewer 1 Report

Comments and Suggestions for Authors

The reviewer appreciates the authors’ effort in revising the manuscript. The authors have mostly addressed the reviewer’s previous concerns and improved the manuscript.

1. Original Comment: Covariates that are potentially predictive of KRAS mutation should be included in the models, especially those significantly different in the internal and external cohorts (e.g., gender, number of liver metastases, median slice thickness).

·       Authors’ response: In literature no predictive covariates of KRAS mutations are described. Moreover, it was decided to firstly assess whether radiomic features had predictive value for determination of KRAS. If the model still had good performance in the external cohort, it could have been considered to include clinical data.

There at least exists divergent evidence that KRAS mutation may be related to age and gender (e.g., PMID23014527 and PMID25073438). Although the reviewer understands that the main aim of the paper is to assess whether radiomic features are good predictors for KRAS mutation, possible confounders should still be included in the statistical modeling.

2. Original Comment: In Table 2, the authors should (1) provide confidence intervals for the model performance metrics and (2) list training set performances in addition to the testing set performances.

·       Authors’ response: We listed the training set performances in section Results Table 2. We do not have the confidence intervals for model performance metrics, as we didn’t apply bootstrapping or cross-validation. We only did cross-validation for tuning of the hyperparameters.

The authors should do so to provide more robust results.

3. Original Comment: Figure 1 (especially the last three columns) should be improved so that the cliparts are less generic and more informative. For instance, the authors may provide concrete examples of different shapes and textures in the extracted features.

·       Response: Figure 1 depicts a schematic and general workflow of the radiomics process. We chose to do so for the readers that are less acquainted with the radiomics workflow. Including different shapes and textures will lead to a less clear figure of the general process. Information on the different features can be found in the Supplementary.

The reviewer understands the authors’ concern about obscuring the main message of the figure, but this is not necessarily the case. The shapes and textures can be presented as (simplified) images as well and will not change the overall framework/complexity of the flow chart. Unfortunately, the current subfigures in the last three columns do not provide any added value to the figure, and “volume of interest” and “shape” used the same cliparts, which can be confusing to the audience.

Reviewer 2 Report

Comments and Suggestions for Authors

The new version of manuscript seems better than previous one.  It's well designed, and good for publication.
